

# AMAS: a fast tool for alignment manipulation and computing of summary statistics

Marek L. Borowiec

Department of Entomology and Nematology, UC Davis, Davis, United States

## ABSTRACT

The amount of data used in phylogenetics has grown explosively in the recent years and many phylogenies are inferred with hundreds or even thousands of loci and many taxa. These modern phylogenomic studies often entail separate analyses of each of the loci in addition to multiple analyses of subsets of genes or concatenated sequences. Computationally efficient tools for handling and computing properties of thousands of single-locus or large concatenated alignments are needed. Here I present AMAS (Alignment Manipulation And Summary), a tool that can be used either as a stand-alone command-line utility or as a Python package. AMAS works on amino acid and nucleotide alignments and combines capabilities of sequence manipulation with a function that calculates basic statistics. The manipulation functions include conversions among popular formats, concatenation, extracting sites and splitting according to a pre-defined partitioning scheme, creation of replicate data sets, and removal of taxa. The statistics calculated include the number of taxa, alignment length, total count of matrix cells, overall number of undetermined characters, percent of missing data, AT and GC contents (for DNA alignments), count and proportion of variable sites, count and proportion of parsimony informative sites, and counts of all characters relevant for a nucleotide or amino acid alphabet. AMAS is particularly suitable for very large alignments with hundreds of taxa and thousands of loci. It is computationally efficient, utilizes parallel processing, and performs better at concatenation than other popular tools. AMAS is a Python 3 program that relies solely on Python's core modules and needs no additional dependencies. AMAS source code and manual can be downloaded from http://github.com/marekborowiec/AMAS/ under GNU General Public License.

## INTRODUCTION

The amount of data used in modern phylogenetics has increased dramatically since the advent of next-generation sequencing (*McCormack et al., 2013*). Data sets composed of hundreds or thousands of loci are becoming commonplace. Efficient concatenation of alignments is important in modern phylogenetics where multiple concatenation procedures are often carried out to explore how phylgoenetic signal is structured across the data. For example, two recent phylogenomic studies (*Sharma et al., 2014*; *Borowiec et al., 2015*) required concatenation and independent inference on dozens of data sets varying e.g., in

Corresponding author
Marek L. Borowiec,
mlborowiec@ucdavis.edu

**Table 1** Overview of functions available in AMAS, FASconCAT-G, and Phyutility.

| Function | AMAS | FASconCAT-G | Phyutility |
|---|---|---|---|
| Input formats | fasta phylip nexus | clustal fasta phylip nexus | fasta nexus |
| Concatenation | yes | yes | yes |
| Splitting or site extraction | yes | yes | yes (gaps only) |
| Summary statistics | yes | yes | no |
| Replicate alignments | yes | no | no |
| Taxon removal | yes | no | no |
| Translation | no | yes | no |
| RY coding | no | yes | no |
| Consensus sequences | no | yes | no |
| NCBI interactions | no | no | yes |
| Tree manipulations | no | no | yes |

rate of evolution or the amount of missing data. Alignment summary statistics are also needed for identification and filtering out "gappy" or fast-evolving data from downstream analyses such as the ones carried out in the studies cited above. Because the size of alignments used in phylogenetics is growing rapidly, there is a need for a fast and easy to use tool that can supplement existing phylogenomic pipelines. Available tools used for concatenation of alignments are either unable to correctly parse all alignment formats, are not flexible in the output format, or are slow to work on very large data sets (see Performance section below). Modern phylogenetic analysis often requires a form of bioinformatics pipeline where output of one procedure is being redirected as input for another tool. Although a number of freely available tools for manipulating alignments and computing their basic statistics exist, some of the most popular ones are based on graphical user interfaces (e.g., Mesquite: *Maddison & Maddison, 2015*) and not appropriate for command-line or scripted pipeline analyses. Examples of command-line tools with functionality partially overlapping with AMAS are FASconCAT-G *Kück & Longo (2014)* and Phyutility *Smith & Dunn (2008)*. Phyutility and FASconCAT-G both allow for concatenation and the latter is additionally capable of computing various alignment statistics. All three programs have a broad range of functions and direct comparisons are difficult (see Table 1 for an overview of functions). More specifically, in addition to conversion, concatenation, and producing alignment summaries, FASconCAT-G allows the user to, among other functions, write MrBayes blocks in NEXUS files or create consensus sequences. Phyutility also allows for interactions with the NCBI databases and a number of manipulations on phylogenetic trees. Because of the many non-overlapping functions of these programs, the comparisons here focus on computing time required for concatenation. It should also be emphasized that the tools compared here should be viewed as complementary and AMAS is not intended to replace them. As demonstrated below, however, AMAS outperforms both FASconCAT-G and Phyutility at concatenation on a single core of a desktop computer. AMAS also supports parallel processing for even faster computation using multiple cores. It is easy to install and use, requires only a standard distribution of Python 3 or newer, and is provided with a detailed instructions manual.

## METHODS

To assess and compare the performance of the concatenation fuctions of AMAS, FASconCAT-G, and Phyutility, I used four recently published phylogenomic data sets: filtered DNA alignments of 8,295 exons of 52 vertebrates and 3,679 UCE loci of 49 amniote taxa *Jarvis et al. (2014)* (available at gigadb.org/dataset/101041), DNA and amino acid alignments of 1,478 loci from 144 arthropod taxa from *Misof et al. (2014)* (http://datadryad.org/resource/doi:10.5061/dryad.3c0f1), and amino acid alignments of 5,214 exons from 19 hymenopteran taxa from *Johnson et al. (2013)* (datadryad.org/resource/doi:10.5061/dryad.jt440).

FASconCAT-G v1.02 (*Kück & Longo, 2014*) was downloaded from https://www.zfmk. de/en/research/research-centres-and-groups/fasconcat-g and used according to the user manual provided along with the software download. Concatenation runs with FASconCAT-G were done in two modes: with the -i option that prevents the program from simultaneous calculation of alignment statistics for faster computing times, and with simultaneous writing of the statistics. The former is comparable with the AMAS concatenation option. Phyutility (*Smith & Dunn, 2008*) is available at https://code.google.com/p/phyutility/. It was used according to the manual. For a complete list of verbatim commands used for benchmarking, see Table S1.

The performance was assessed on a desktop computer equipped with 12 Intel(R) Core(TM) i7-4930K CPUs at 3.40 GHz, 32 GB of DDR3 RAM at 1,600 Hz, a generic 4TB 7,200 rpm hard drive, and a Ubuntu 12.04LTS operating system with Python 3.4.3 installed.

I used the Unix command time -p to evaluate execution times. Each command was run three times with minimal background system load and average time recorded.

## AMAS ALGORITHMS

When used from the command line, AMAS parses input files along with associated metadata into taxon names and sequences stored as basic Python object types such as strings, lists, dictionaries, and tuples. Summary calculations or manipulations such as concatenation or taxon removal are then performed on those objects. Following any action requested by the user, AMAS composes strings in appropriate, requested format and writes output into files. A visual explanation of several AMAS functions is provided in Fig. 1. Within the AMAS source code, most of the functionality is wrapped into the custom MetaAlignment class that allows the user to interact with the data using the program as a Python module, in addition to the command line. Using the MetaAlignment methods the user can interact directly with basic alignment meta data and components such as file names, taxa, and constituent sequences using tools for correct parsing of input and output formatting.

## FUNCTIONALITY AND USAGE

The open source code of AMAS and its manual are available at http://github.com/marekborowiec/AMAS/ or at Python's Package index at https://pypi.python.org/pypi/amas/. AMAS has been tested on a number of data sets to assure that all supported

## A: concatenation

`AMAS concat -i FILE_1.fas FILE_2.fas -f fasta -d dna --concat-out CONCATENATED_FILE.fas`

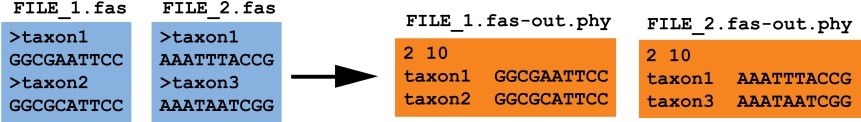

## B: format conversion

`AMAS convert -i FILE_1.fas FILE_2.fas -f fasta -d dna --out-format phylip`

## C: alignment splitting

`AMAS split -i CONCATENATED_FILE.fas -f fasta -d dna --split-by partitions.txt --remove-empty`

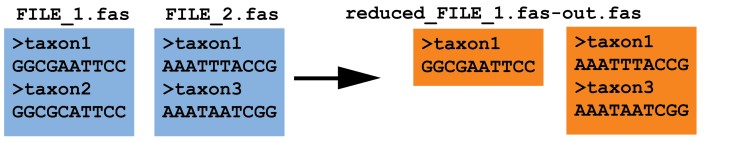

## D: taxon removal

`AMAS remove -i FILE_1.fas FILE_2.fas -f fasta -d dna -x taxon2`

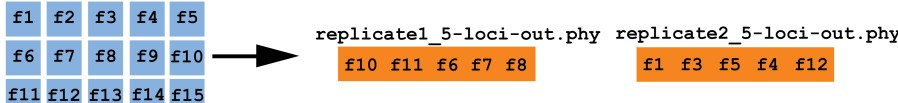

## E: creating random replicate alignments

`AMAS replicate -i f*.phy -f phylip -d aa -r 2 5 --out-format phylip`

**Figure 1  AMAS functionality.** (A) Concatenation of two FASTA files. (B) File format conversion from FASTA to PHYLIP. (C) Splitting of a concatenated alignment according to pre-defined partitions. (D) Removal of sequences by name. (E) Creation of randomized replicates from input alignments. All input files are preserved, here indicated in blue. Orange files represent the output. Command line examples are given along with each action.

formats are correctly parsed. It is a project under active development and, following user suggestions, additional features are likely to be implemented in the future.

## Amas as a Python package

AMAS uses the custom `MetaAlignment` class and its methods to handle multiple sequence alignments. All the major functions available on the command line are available through this interface but additional flexibility is added by being able to directly manipulate parsed sequences and use built-in methods. A discussion of available methods is available in the manual supplied with the package. I focus on the command line options when discussing AMAS capabilities below.

## Command line interface

AMAS requires the user to supply information on the alignment: (1) the name of input file(s), (2) the input format, and (3) whether it contains nucleotide or amino acid sequences. The user also needs to specify one action to be performed by the program. These actions are explained below. AMAS uses Python's `argparse` module to handle user-provided arguments (or flags) and is agnostic of the order in which they are given. Correct specification of the input format is crucial for correct parsing of the files. While AMAS takes minimal measures to detect if the right format was given, there is a trade-off between computation time and automated detection of file formats. Detailed usage instructions are provided in the instruction manual and example input/output is visualized in Fig. 1.

AMAS allows easy conversion of formats (Fig. 1B) for thousands of files in seconds. It is able to parse five of the most popular formats of multiple sequence alignments: FASTA, PHYLIP (both sequential and interleaved formats are supported), and NEXUS (sequential and interleaved).

In addition to conversion, AMAS is fast at concatenation (Fig. 1A) and the input and output files can be in any of the supported formats. It also creates a partitions file that records the coordinates of each locus in the concatenated alignment. AMAS is competitive relative to other popular software used for this purpose: see Performance section below.

AMAS allows the user to create concatenated alignments from a number of randomly chosen alignments that can be used for, for example, the phylogenetic jackknife (Fig. 1E). Assuming a large set of input files (e.g., thousands of single-locus alignments or arbitrary sets of sites created by splitting with AMAS; see below), one can create any number of replicate alignments, each concatenated from any number of files randomly chosen from the input set.

AMAS allows writing files from input alignments, given a list of sites (Fig. 1C). This can be used to split alignments by partition in matrices where the data originally supplied is only in the form of concatenated matrix but with information on partitioning schemes present.

It is also easy to trim taxa from alignments using AMAS (Fig. 1D).

Finally, AMAS can calculate alignment properties that include the number of taxa, alignment length, total number of matrix cells, overall number of undetermined characters, percent of missing data, AT and GC contents (for DNA alignments), number and proportion of variable sites, number and proportion of parsimony informative sites,

**Table 2  Select statistics of benchmark alignments.**

| Alignment name | Data type | Length | Total cells | Missing percent | Prop. variable | Prop. parsimony inf. |
|---|---|---|---|---|---|---|
| *Johnson et al. (2013)* | Amino acid | 3,001,657 | 57,031,483 | 46.42 | 0.53 | 0.28 |
| *Misof et al. (2014)* | Amino acid | 1,313,129 | 189,090,576 | 64.43 | 0.76 | 0.59 |
| *Jarvis et al. (2014)* UCEs | Nucleotide | 9,251,694 | 453,333,006 | 19.46 | 0.64 | 0.44 |
| *Jarvis et al. (2014)* exons | Nucleotide | 13,557,123 | 704,970,396 | 16.63 | 0.51 | 0.35 |

and counts of all characters in a given amino acid or nucleotide alphabet. An example of summary output for data sets used here for benchmarks can be found in Table S1.

## PERFORMANCE

I assessed the performance of AMAS on four alignments ranging from, when concatenated, approximately 57 to 705 million matrix cells in total and occupying ca. 56 to 681 MB of hard drive space as FASTA files. The alignment length in total number of sites (amino acids or DNA bases), number of matrix cells, percent of missing data, and proportion of parsimony-informative sites, as calculated by AMAS, are given in Table 2.

### Conversion among formats

Conversion among formats is fast with AMAS. The data set of 5,214 alignment files from *Johnson et al. (2013)* was converted among all formats in around 3 s on a single core and 1.5 using all 12 cores (see Table S2).

### Concatenation

Concatenation is a function that can be performed by two other popular programs that are used for alignment manipulations in phylogenomic data sets: FASconCAT-G, a Perl program (*Kück & Longo, 2014*) and Phyutility, written in Java (*Smith & Dunn, 2008*). The former supports FASTA, PHYLIP, and CLUSTAL alignments as input, the latter only FASTA and NEXUS formats. Here I compare the performance of both to AMAS. FASconCAT-G worked on the amino acid alignments of *Johnson et al. (2013)* and *Misof et al. (2014)* data sets, concatenating the former in over 36 s and the latter in over 250 s. This program also allows simultaneous computation of alignment summaries during concatenation. This took about 8 and 23 min for the two data sets, respectively. FASconCAT-G crashed with an error when attempting to read in the nucleotide FASTA files of *Jarvis et al. (2014)* UCE and exon data sets. Phyutility was able to concatenate all of the four FASTA data sets, in times of about 10 min for the smallest data set of 5,214 amino acid loci of *Johnson et al. (2013)* to over 3 h for the largest data set of 8,295 exons from *Jarvis et al. (2014)*. Phyutility allows concatenation only to the sequential NEXUS format. AMAS concatenation times ranged from less than 2 s for the smallest data set of *Johnson et al. (2013)* to under 18 s for *Jarvis et al. (2014)* data (and only 1 to 8 s, respectively, using all 12 cores; see Table S2), outperforming the other two programs by at least a factor of 30. A comparison of times taken for concatenating the smallest data set of 5,214 amino acid loci from *Johnson et al. (2013)* is presented in Fig. 2A. See Table S2 for additional

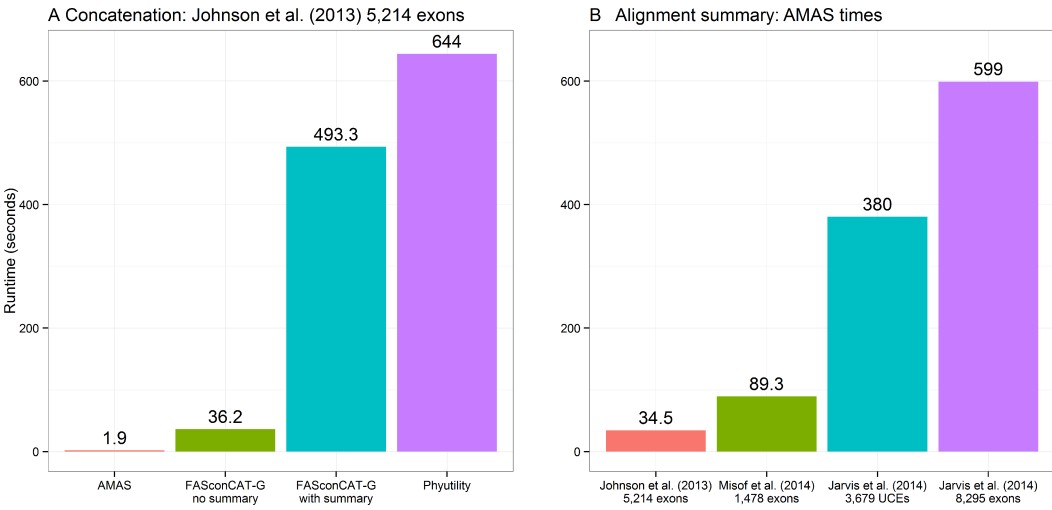

**Figure 2  Performance.** (A) Computing times for concatenation of the *Johnson et al. (2013)* data set composed of 5,214 separate alignments. FASconCAT-G was run in two modes: with and without simultaneous computation of alignment summaries (`FASconCAT-G v1.02.pl -s` and `FASconCAT-G v1.02.pl -s -i`, respectively). Phyutility was ran with `java -jar phyutility.jar -concat -in *fas -out phyut j2013-test`. (B) Computing times for AMAS writing alignment summaries on the four benchmark data sets (`AMAS.py summary` command).

comparisons, where commands and data used for Fig. 2 is indicated by the '-fig' suffix under 'Function' column.

## Computing summary statistics

The times required for computing summaries are similar regardless of whether the data sets are processed as many single-locus alignments or a concatenated matrix and I present the results on concatenated alignments only (Fig. 2B). These are also similar regardless of the input file format. See Table S2 for an exhaustive list of benchmarks. FASconCAT-G (*Kück & Longo, 2014*) also calculates alignment statistics but it is significantly slower than AMAS. The total time taken for FASconCAT-G for concatenation and summary computing on the smallest data set of *Johnson et al. (2013)* was >500 s, while AMAS concatenates this data set in under 2 s and outputs summaries in another 35 s (Fig. 2).

## Splitting by partition

The process of splitting concatenated files is also very fast. AMAS wrote 5,214 files from partitions of the *Johnson et al. (2013)* concatenated matrix in about 18 s.

## DISCUSSION

AMAS is a fast program, suitable for simple manipulations useful in phylogenetic inference. It is robust to various input data formats and outperforms other popular programs at concatenation without any cost in accuracy. It is also potentially much more flexible because of its design as a Python package. Future improvements to AMAS are anticipated, including sequence recoding, translation, and adding functions to manipulate the alignments on a by-taxon basis.

The biggest advantage of AMAS is the very fast processing of input that can be performed through command line interface as well as a Python module import. It is user friendly and provides flexibility as a Python package for the experienced user, at the same time being useful for a person without scripting experience or someone simply wishing to perform basic operations quickly from the command line. While built-in module functions of AMAS are not as extensive as those available in more sophisticated Python libraries such as Biopython (*Cock et al., 2009*), focus specifically on aligned sequences may be appealing: Access to raw alignment components such as sequences and taxa is easy in AMAS, achieved without learning potentially overwhelming complexity of classes and methods available in other packages.

The continuing growth in the amount of sequence data used in phylogenetics may require that even faster similar tools be developed soon, taking advantage of compiled languages such as C++ or Julia. At present, however, AMAS offers a tool that is available to any potential user with access to a command line interface, allowing fast computing on some of the largest alignments published to date.

## ACKNOWLEDGEMENTS

I would like to thank Carlos Peña for help in the initial stages of this project and Brian Johnson for access to the test desktop computer. Many thanks to Ernie Lee and Joanna Chiu for valuable comments on program's interface and documentation. Three anonymous reviewers, the editor, and Phil Ward provided comments that helped to improve this manuscript.

### Funding

The author received support from the NSF Doctoral Dissertation Improvement Grant DEB-1402432 and a grant from Microsoft Azure Research. The funders had no role in study design, data collection and analysis, decision to publish, or preparation of the manuscript.

### Competing Interests

The author declares there is no competing interests.

### Author Contributions

- Marek L. Borowiec conceived and designed the experiments, performed the experiments, analyzed the data, contributed reagents/materials/analysis tools, wrote the paper, prepared figures and/or tables, reviewed drafts of the paper.

### Data Availability

Github: https://github.com/marekborowiec/AMAS/

Dryad: DOI:10.5061/dryad.p2q52.

## Supplemental Information

Supplemental information for this article can be found online at http://dx.doi.org/10.7717/peerj.1660#supplemental-information.

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
