# Peer review of "AMAS: a fast tool for alignment manipulation and computing of summary statistics"

_PeerJ, doi:10.7717/peerj.1660_

## Round 0.1 · original submission · Major Revisions

Please address the following concerns raised by the reviewers.

First, all three reviewers pointed out that there is no comparison of AMAS with existing tools (e.g., FASconCAT-G and Phyutility). Besides the computational time, please compare the results of AMAS with those of existing tools in terms of their accuracy. Also, please describe the advantages and disadvantages of AMAS, and why it will serve as a useful new tool to the phylogenetic field.

Second, please add a diagram to illustrate how the AMAS method works. More texts are also need to explain the method in greater detail.

Reviewer 1 ·

Basic reporting

I have two major concerns:

1. It is still unclear to what concatenation can do and why reducing the concatenation time is important. Some explanations or a mini example would be helpful for us to understand.

2. It would be great to have some examples of summary statistics from one of the datasets mentioned, so readers could better understand the output of the algorithm.

Experimental design

From the description in the main text, there are some similar tools that can do the same job as AMAS (at least in concatenation), including FASconCAT-G, Phyutility. The author demonstrated that AMAS is much faster (in Figure 1), but there’s no comparison of how good the output of each program is. Some evaluations on the results of these programs should be added.

Validity of the findings

No Comments.

Additional comments

In this paper, the author presented an open source tool, AMAS, for fast alignment and summary statistics generation of phylogenesis studies. Although I’m not an expert on phylogeny, it does seem that AMAS is a very useful tool and it greatly reduce the time needed for concatenation and alignment summary generation. While it seems that AMAS would be very helpful for phylogenesis researchers, I have a few concerns that are already addressed in the three review criterias mentioned above. Some other minor comments include:

— It would be great if there is a diagram of components of AMAS, including input/output files, and the connection between different AMAS components.

— The type of AMAS software license (BSD, MIT, etc.) should be specified.

Reviewer 2 ·

Basic reporting

No Comments

Experimental design

No Comments

Validity of the findings

There was no comparison of results themselves from AMAS and known software (FASconCAT-G and Phyutility). It would help to see whether the impressive improvement in computational speed has any cost associated with the accuracy of the results, if the author can show the differences (or lack of them) between AMAS and known software.

Reviewer 3 ·

Basic reporting

1. Datasets used for comparison would be better if they were organized in a supplementary table, instead of listed in the main text.
2. Detailed usage was included in the main text, which is unnecessary, since the manual for github already provided usage information. General description of functions would be enough.

Experimental design

1. The introduction included the purpose of AMAS, but the difference of AMAS with other applications may need to be emphasized more. For example, whether the speed of current applications is not enough for research purpose, or why functions provided by AMAS is useful for research compared with existing tools.

Validity of the findings

1. One of the major advantages of AMAS is the speed. But for the running time comparison, only the best one was used, which might be biased by system load or other factors. So at least three times running should be reported with error bar.
2. The author failed to compare results of AMAS with existing tools, so we are not sure if the AMAS is faster since it provides incorrect result.
3. It is unclear if the other tools during comparison provided other outputs, so the time may not be comparable.

---

## Round 0.2 · Minor Revisions

Please refer to Reviewer 3's comments for the final minor revisions.

Reviewer 1 ·

Basic reporting

The revision is satisfactory and I have no further comments.

Experimental design

No comments

Validity of the findings

No comments

Additional comments

No comments

Reviewer 2 ·

Basic reporting

No Comments

Experimental design

No Comments

Validity of the findings

No Comments

Additional comments

The author has provided an overview of functions of AMAS vs published tools (FASconCAT-G and Phyutility) and shown the rationale of having a very fast algorithm for concatenation while getting the correct end results. Therefore, I would recommend this manuscript be accepted as is.

Reviewer 3 ·

Basic reporting

The updated version is clear and well-written, while there are still some minor problems to fix. For example, Table1 should be Table2 in the text "as calculated by AMAS, are given in Table1.".

Experimental design

The overall experiment is rigorous. The methods are described clearly. But the command used for two other tools in Figure2A need to be specified to reproduce the result.

Validity of the findings

The data is robust. But it would be great if the error bars are used in Figure 2A.

Additional comments

The AMAS seems to be a useful tool in phylogenomic studies, especially for large datasets.

---

## Round 0.3 · accepted · Accept

The current figures are in .png format. If our production team thinks higher resolution figures are needed, you may need to replace them.